# Solving a Simple Geduldspiele Cube with a Robotic Gripper via Sim-to-Real Transfer

Ji-Hyeon Yoo [1], Ho-Jin Jung [1], Jang-Hyeon Kim [1], Dae-Han Sim [1] and Han-Ul Yoon [1,2,*]

1   Department of Computer Science, Yonsei University, Wonju 26493, Korea
2   Division of Software, Yonsei University, Wonju 26493, Korea
*   Correspondence: huyoon@yonsei.ac.kr; Tel.: +82-33-760-2235

**Abstract:** Geduldspiele cubes (also known as patience cubes in English) are interesting problems to solve with robotic systems on the basis of machine learning approaches. Generally, highly dexterous hand and finger movement is required to solve them. In this paper, we propose a reinforcement-learning-based approach to solve simple geduldspiele cubes of a flat plane, a convex plane, and a concave plane. The key idea of the proposed approach is that we adopt a sim-to-real framework in which a robotic agent is virtually trained in simulation environment based on reinforcement learning, then the virtually trained robotic agent is deployed into a physical robotic system and evaluated for tasks in the real world. We developed a test bed which consists of a dual-arm robot with a patience cube in a gripper and the virtual avatar system to be trained in the simulation world. The experimental results showed that the virtually trained robotic agent was able to solve simple patience cubes in the real world as well. Based on the results, we could expect to solve the more complex patience cubes by augmenting the proposed approach with versatile reinforcement learning algorithms.

**Keywords:** geduldspiele cube; robotic agent; sim-to-real transfer; reinforcement learning; proximal policy optimization

## 1. Introduction

A reinforcement learning (RL) algorithm solves a problem by finding the optimal policy through exploitation and exploration for a large number of episodes [1,2]. During this iterative process, the RL algorithm gathers sampling-based statistics, the expected cost/reward, after performing an action, which in turn serves as a basis to infer a cost/reward functional. From an algorithmic viewpoint, the RL is also a class of data-driven approaches to solve a sequential decision-making process given a model-free situation; therefore, a "the more, the better" rule still holds for the size of datasets [3–5]. For physical systems in practice, however, running the experiment through the large number of episodes is somewhat impossible due to the fatigue of mechanical parts as well as safety-related issues, e.g., potential risk, unexpected behavior, and so on [6].

"Sim-to-real transfer" is one way to overcome the abovementioned problematic issues in RL. The key idea of sim-to-real transfer is to train a machine learning (ML) agent in a simulation-based environment and deploy the virtually trained ML agent in the real world. For a decade, there have been promising results disseminated in a broad range of robotic application fields, such as peg-in-hole [7–9], cabinet door closing [7,9], handling deformable object [10], complex dexterous manipulation [11], indoor visual navigation [12,13], and so on. For machine-vision-involved robotic tasks, e.g., pick-and-place, vision-based grasping, and deep visual servoing, domain adaptation/randomization has been applied to increase the variety of domain and enhance task performance [14–17].

Recalling that the sim-to-real transfer aims at solving the real-world RL problems ultimately, we can consider two implementation directions as follows:

- A simulation world guarantees a physics engine with superior fidelity in order to deploy a virtually trained RL agent into the real world directly after training.
- A small discrepancy in sense of physics between the simulation world and the real world is allowed; however, the RL agent is endowed with the ability to overcome/adapt the discrepancy that was not experienced during the simulation-based training.

The former direction corresponds to the recent trend to develop more realistic physics engines, such as MuJoCo [18], Raisim [19,20], Algoryx [21], Airsim [22], CARLA [23], RotorS [24], etc. In contrast, the latter one is well exemplified by the meta-learning approach [16,25] and policy distillation [26,27]. Of course, these two directions have their own pros and cons. Nevertheless, developing the physics model with superior fidelity would rather be difficult for tasks involving haptic interactions, i.e., grasping a deformable object. In addition, if we have perfect knowledge about a system model, including uncertainties, then it would be better to apply a traditional control-based approach, e.g., a decentralized control method to control a two-interconnected inverted pendulum [28,29].

A human being is a fast learner who can learn with few data indeed [25,30]. In our daily living activities, dexterous hand manipulation, e.g., opening a bottle cap, twisting door knobs, using chopsticks, is the best example showing the human ability to learn complex tasks [31]. Under the RL framework, the optimal policy imitating human behavior can be derived [32,33], and an inverse reinforcement learning approach enables us to model a human behavior as a value function [34,35]. Compared to the RL agent, nonetheless, the human being uses a meta-heuristic approach when the given task is highly complex, such as dexterous hand manipulation [32,36]. Existing studies have reported successful results for placing and pushing [33,37], door-locking, and Jenga [38]; however, the feasibility of the RL algorithm for learning human-like meta-heuristic behavior in dexterous hand manipulation has not yet been fully accounted for.

Inspired by the aforementioned findings to develop a dexterous robotic manipulation system, in this paper, we propose an approach to solve the geduldspiele cubes via sim-to-real transfer. Specifically, we first developed a dual-arm robot to manipulate the geduldspiele cubes by a wrist–hand action in the real world, as well as a virtual robotic agent in the simulation world, which serves as the avatar of the real robotic system. Next, we trained the virtual robotic agent by proximal policy optimization under reinforcement learning framework and scrutinized the feasibility of solving the more complex geduldspiele cube in both the simulation world and the real world. Finally, the experimental results showed that the dual arm controlled by the virtually trained robotic agent could solve the given geduldspiele cubes via the proposed sim-to-real transfer. Throughout this study, our research question was "Can the virtually trained RL agent learn human-like optimal policy to solve a simple geduldspiele cube?".

The rest of the paper is organized as follows: our approach to solve the geduldspiele cubes via sim-to-real transfer is introduced in Section 2. Specifically, our approach to cube designs, dynamic models, and a developed sim-to real transfer framework are explained. In Section 3, the experiment to train and evaluate the robotic agent for both the simulation world problem and the real world problem is presented. The results are reported and discussed in Section 4. Section 5 will be the conclusion and future work of this paper.

## 2. Approach to Solving Geduldspiele Cubes via Sim-to-Real Transfer

### 2.1. Geduldspiele Cubes in a Robotic Gripper: Flat, Convex, and Concave

Figure 1 shows a geduldspiele cube in a robotic gripper (left column) and the three different types (right column) which are considered throughout this paper. In Figure 1, those in the top row are the robotic hand and geduldspiele cubes in the real world and those in the bottom row are objects in the simulation created by Blender (version 3.0.0, Blender Foundation). The curvatures of a plane in each cube are flat, convex, and concave from the left to the right, respectively.

For the geduldspiele cubes, the given task is to locate an iron ball on a center hole so that the iron ball is stuck to the center hole and does not move. In fact, there exist 24 different types of geduldspiele cubes for which highly dexterous hand manipulation, as well as patience, is required to complete the cubes. Since the main purpose of this study is to scrutinize the feasibility of solving the geduldspiele cubes by sim-to-real transfer, wherein a reinforcement learning will serve as a playground to virtually train a robotic agent, we start by challenging rather simpler ones.

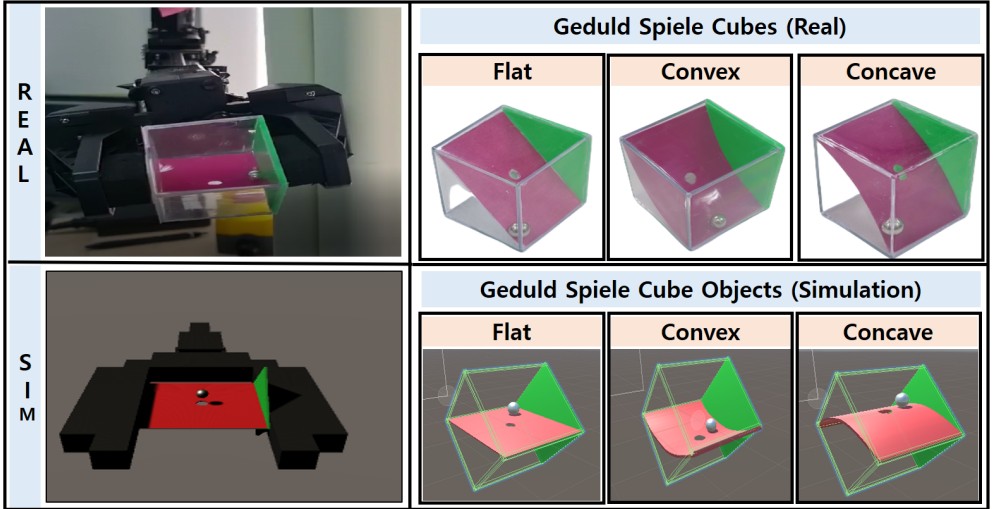

**Figure 1.** A geduldspiele cube in a robotic gripper and the three different cube types according to the curvature of a plane in the cube: the robotic gripper and the cubes in the real world (**top row**) and the corresponding objects in the simulation (**bottom row**).

### 2.2. Dynamic Models: Ball–Plane Model and Ball–Hole Model

To derive the dynamic models, we hold the following two assumptions:

- The plane of the geduldspiele is flat.
- The linear velocity of the iron ball always passes through the center of the hole.

Our main purpose of analyzing dynamic models is to obtain an insight to define a state vector and an action to virtually train our robotic agent in a reinforcement learning framework. Indeed, Singh and Sutton introduced a mountain car problem solution approach in which the slope is considered as various frictions along the $x$-axis [39]. On the basis of their findings, therefore, we also expect that the virtually trained robotic agent can learn to solve the aforementioned geduldspiele cubes, which are flat, convex, and concave, by regarding the curvature of the plane as a friction-like effect.

Figure 2 shows the robotic gripper and plane coordinate, wrist joint actuators, and two ball–plane models with respect to rotating axes. We set the plane coordinates so that $x$-axis and $y$-axis are aligned to rotation axes $z_4$ and $z$-axis, respectively, as illustrated in Figure 2a. Consequently, the $xy$-plane is aligned to the plane of the geduldspiele cube; hence, the iron ball will be moving by tilting the plane with two actuators, which are denoted by A and B. For this robotic gripper and geduldspiele cube system, the ball–plane model and ball–hole model are discussed below. Note that we introduce the two models by recapitulating the exiting models presented in [40–42].

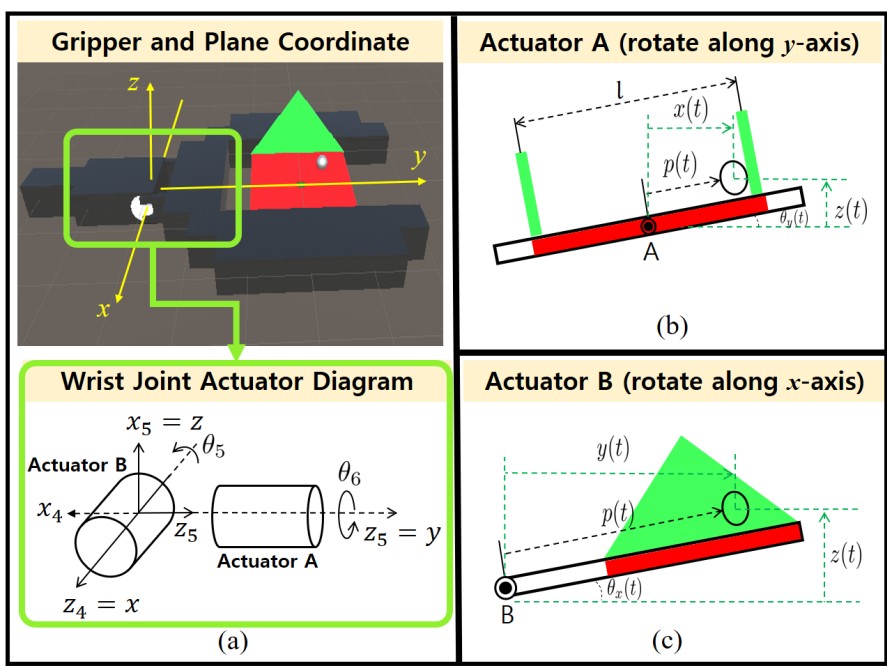

**Figure 2.** Coordinate setting: (**a**) the robotic gripper and plane coordinate and wrist joint actuators, (**b**) actuator A and corresponding iron ball action along *y*-axis, and (**c**) actuator B and corresponding iron ball action along *x*-axis.

### 2.2.1. Ball–Plane Model

By following [40,41], let $K_p$ and $K_b$ be the kinetic energy of the plane and the iron ball, respectively. $K_p$ and $K_b$ can be defined as

$$K_p = \frac{1}{2}J_p\dot{\theta}^2$$
$$K_b = \frac{1}{2}J_b\dot{\beta}^2 + \frac{1}{2}m_b v_b^2 \tag{1}$$

where $J_p$ is the moment of inertia of the plane and $\dot{\theta}$ is the angular velocity of the plane. For the iron ball, $J_b$, $\dot{\beta}$, $m_b$, and $v_b$ represent the moment of inertia, the angular velocity, the mass, and the linear velocity, respectively. From Figure 2b, we can easily find that $\dot{\beta}$ and $v_b$ can be expressed by

$$\dot{\beta} = \frac{\dot{p}}{r_b} \quad \text{and} \quad v_b^2 = \dot{p}^2 + p^2\dot{\theta}^2 \tag{2}$$

where $r_b$ is the radius of the iron ball. Substituting (2) into $K_b$ in (1) yields

$$K_b = \frac{1}{2}\left(\frac{J_b}{r_b^2} + m_b\right)\dot{p}^2 + \frac{1}{2}m_b p^2\dot{\theta}^2 \tag{3}$$

Let $U_p$ and $U_b$ be the potential energy of the plane and the iron ball, respectively. $U_p$ and $U_b$ are defined by

$$U_p = m_p g\frac{l}{2}(1 + \sin\theta)$$
$$U_b = m_b g p \sin\theta \tag{4}$$

where $l$ and $m_p$ are the length and the mass of the plane, and $g$ is a gravitational constant. Now we can define the Lagrangian $L$ for the ball–plane system:

$$L = (K_p + K_b) - (U_p + U_b)$$
$$= \frac{1}{2}\left(\frac{J_b}{r_b^2} + m_b\right)\dot{p}^2 + \frac{1}{2}(J_p + m_b)p^2\dot{\theta}^2 - m_p g \frac{l}{2}\sin(1 + \theta) - m_b g p \sin\theta \tag{5}$$

Let the generalized coordinate be $q(t) = [p(t), \theta(t)]^T$. By the Euler–Lagrange equation of the first kind,

$$\frac{d}{dt}\left(\frac{\partial L}{\partial \dot{p}}\right) - \frac{\partial L}{\partial p} = F_\mu, \tag{6}$$

hence, we have

$$\left(\frac{J_b}{r_b^2} + m_b\right)\ddot{p} + m_b g \sin\theta - m_b p\dot{\theta}^2 = F_\mu \tag{7}$$

where $F_\mu$ is the friction force defined by

$$F_\mu = -\mu N \text{sign}(\dot{p}) = -\mu m_b g \cos\theta \text{sign}(\dot{p}). \tag{8}$$

Similarly, by the Euler–Lagrange equation of the second kind,

$$\frac{d}{dt}\left(\frac{\partial L}{\partial \dot{\theta}}\right) - \frac{\partial L}{\partial \theta} = \tau, \tag{9}$$

we obtain

$$(m_b p^2 + J_p)\ddot{\theta} + 2m_b p\dot{p}\dot{\theta} + m_b g p \cos\theta + m_p g \frac{l}{2}\cos\theta = \tau \tag{10}$$

where $\tau$ is a torque generated by the actuator.

### 2.2.2. Ball–Hole Model

Figure 3 depicts the motion of an iron ball after crossing a closer boundary of a hole. In Figure 3a, the iron ball is approaching the other side of the hole. $\dot{\beta}^-$ represents the angular velocity of the iron ball just before hitting the other side, and $v_{bt}^-$ and $v_{br}^-$ are the tangential and radial component of $v_b$, respectively. Those three values just after hitting the other side are represented by $\dot{\beta}^+$, $v_{bt}^+$, and $v_{br}^+$, as shown in Figure 3b. $\phi$ is the angle between a line connecting a ball center to a contact edge point and the vertical. Figure 3c shows that the iron ball is at rest.

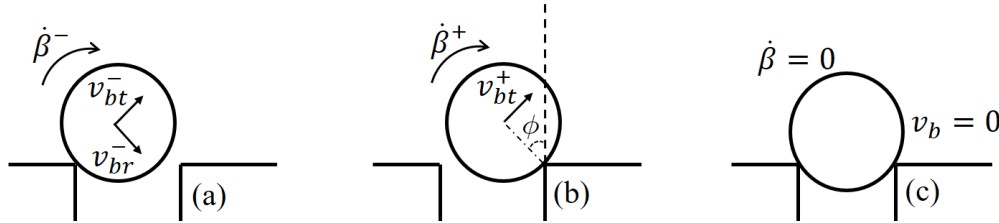

**Figure 3.** The motion of the iron ball (**a**) just before and (**b**) just after hitting the boundary of the hole, and (**c**) the iron ball at rest (this figure is redrawn from [42]).

According to [42], there exist three possibilities for an iron ball after crossing the boundary of a hole:

(1)  $v_{bt}^+$ might be negative, which implies the iron ball will be at the rest position eventually.

(2)   $v_{bt}^+$ might be positive but small, so that it starts to deviate from the hole (this happens when $\frac{v_{bt}^+}{r_b} < g \cos \phi$). If it does not have enough mechanical energy to surely deviate, then

$$\frac{m_b (v_{bt}^+)^2}{2} + \frac{J_b (\dot{\beta}^+)^2}{2} < m_b g r_b (1 - \cos \phi), \tag{11}$$

and the iron ball will again be at rest.

(3)   $v_{bt}^+$ might be positive and large enough to deviate from the hole (this happens if $\frac{v_{bt}^+}{r_b} > g \cos \phi$ ). Afterward, the iron ball will (i) entirely escape from the hole or (ii) hit the boundary of the hole again.

More elaborated models for ball–hole action can also be found in [43,44].

*2.3. Derivation of LTI Ball–Plane System Model*

By rewriting (7) and (10) as a standard form, we have

$$
\begin{aligned}
\ddot{p} + \frac{m_b g}{J_b / r_b^2 + m_b} \sin \theta - \frac{m_b p}{J_b / r_b^2 + m_b} \dot{\theta}^2 &= \frac{F_\mu}{J_b / r_b^2 + m_b} \\
\ddot{\theta} + \frac{2 m_b p \dot{p}}{m_b p^2 + J_p} \dot{\theta} + \left( \frac{m_b g p + m_p g \frac{l}{2}}{m_b p^2 + J_p} \right) \cos \theta &= \tau
\end{aligned}
\tag{12}
$$

We linearize (12) by analyzing at the equilibrium, say $s_e = [p, \dot{p}, \theta, \dot{\theta}]^T = [0, 0, 0, 0]^T$, which yields a small angle approximation assumption

$$\theta \approx 0, \quad \sin \theta \approx \theta, \quad \cos \theta \approx 1 - \frac{\theta^2}{2} \approx 1.$$

Consequently, we obtain

$$
\begin{aligned}
\ddot{p} &= -\frac{m_b g}{J_b / r_b^2 + m_b} + \frac{F_\mu}{J_b / r_b^2 + m_b} \\
\ddot{\theta} &= -\frac{m_p g l}{2 J_p} + \frac{1}{J_p} \tau
\end{aligned}
\tag{13}
$$

For clarity, we set

$$\frac{1}{J_p} \tau = \frac{m_p g l}{2 J_p} + \bar{\tau}$$

which involves a gravity compensation plus a pure control. Then, (13) becomes

$$
\begin{aligned}
\ddot{p} &= -\frac{m_b g}{J_b / r_b^2 + m_b} + \frac{F_\mu}{J_b / r_b^2 + m_b} \\
\ddot{\theta} &= \bar{\tau}.
\end{aligned}
\tag{14}
$$

Now, let $s = [p, \dot{p}, \theta, \dot{\theta}]^T$; (14) can be expressed as

$$
\begin{aligned}
\dot{s} &= \begin{bmatrix} 0 & 1 & 0 & 0 \\ 0 & 0 & -\frac{m_b g}{J_b / r_b + m_b} & 0 \\ 0 & 0 & 0 & 1 \\ 0 & 0 & 0 & 0 \end{bmatrix} s + \begin{bmatrix} 0 \\ 0 \\ 0 \\ 1 \end{bmatrix} \bar{\tau} + \begin{bmatrix} 0 \\ \frac{F_\mu}{J_r / r_b + m_b} \\ 0 \\ 0 \end{bmatrix} \\
&= As + b\bar{\tau}
\end{aligned}
\tag{15}
$$

By checking the controllability matrix, e.g., `ctrb(A,b)` in MATLAB, we can easily know that the above system is controllable. Given desired eigenvalues $\lambda_i$, we can find a state feedback control $\bar{\tau} = -ks$ yielding

$$\dot{s} = (A - bk)s \tag{16}$$

which leads $A - bk$ to be Hurwitz matrix and is stable.

Recall that the geduldspiele cube is controlled by two actuators along with the *x*- and *y*-axis. By restoring $p = [x, y]^T$, $\theta = [\theta_x, \theta_y]^T$, and $\bar{\tau} = [\bar{\tau}_x, \bar{\tau}_y]^T$, we have a continuous-time state-space representation as follows:

$$s = [x, \dot{x}, \theta_x, \dot{\theta}_x, y, \dot{y}, \theta_y, \dot{\theta}_y]^T \tag{17}$$

and

$$\dot{s} = \begin{bmatrix} A_{11} & A_{12} \\ A_{21} & A_{22} \end{bmatrix} s + \begin{bmatrix} b_{11} & b_{12} \\ b_{21} & b_{22} \end{bmatrix} \bar{\tau} + \bar{F}_\mu \tag{18}$$

where

$$A_{11} = A_{22} = \begin{bmatrix} 0 & 1 & 0 & 0 \\ 0 & 0 & -\frac{m_b g}{J_b/r_b + m_b} & 0 \\ 0 & 0 & 0 & 1 \\ 0 & 0 & 0 & 0 \end{bmatrix}, \quad A_{12} = A_{21} = \begin{bmatrix} 0 & 0 & 0 & 0 \\ 0 & 0 & 0 & 0 \\ 0 & 0 & 0 & 0 \\ 0 & 0 & 0 & 0 \end{bmatrix},$$

$$b_{11} = b_{22} = [0, 0, 0, 1]^T, \quad b_{12} = b_{21} = [0, 0, 0, 0]^T,$$

and

$$\begin{aligned} \bar{F}_\mu &= \frac{1}{J_r/r_b + m_b} [0, F_{\mu,x}, 0, 0, 0, F_{\mu,y}, 0, 0]^T \\ &= \frac{1}{J_r/r_b + m_b} [0, -\mu m_b g \cos\theta_x \text{sign}(\dot{x}), 0, 0, 0, -\mu m_b g \cos\theta_y \text{sign}(\dot{y}), 0, 0]^T \end{aligned}$$

In addition, the state feedback control has the form of

$$\bar{\tau} = \begin{bmatrix} \bar{\tau}_x \\ \bar{\tau}_y \end{bmatrix} = \begin{bmatrix} k_1 & k_2 & k_3 & k_4 & 0 & 0 & 0 & 0 \\ 0 & 0 & 0 & 0 & k_5 & k_6 & k_7 & k_8 \end{bmatrix} s \tag{19}$$

where $k_1$ through $k_8$ can be calculated from given desired eigenvalues.

By rewriting (18) with shorthand notation, we have

$$\dot{s} = As + b\bar{\tau} + \bar{F}_\mu \tag{20}$$

Now, we can express the corresponding discrete-time state-space representation of (20) as

$$s_{k+1} = (I + A\Delta t)s_k + b\bar{\tau}\Delta t + \bar{F}_\mu \Delta t \tag{21}$$

where $I \in \mathbb{R}^{8 \times 8}$ and $\Delta t$ is a sampling time. We note that the definition of parameters and their values appearing throughout the paper are summarized in Table 1.

**Table 1.** The definition of variables and parameters.

| Name | Symbol | Value | Unit |
|---|---|---|---|
| Ball mass | $m_b$ | $1.057 \times 10^{-3}$ | kg |
| Ball radius | $r_b$ | $3.175 \times 10^{-3}$ | m |
| The moment of inertia of the ball | $J_b$ | $4.262 \times 10^{-9}$ | $kg \cdot m^2$ |
| Plate mass | $l$ | $10 \times 10^{-3}$ | kg |
| Plate length | $l_x$ and $l_y$ | $35.73 \times 10^{-3}$ and $51.14 \times 10^{-3}$ | m |
| The moment of inertia of the plate | $J_{p,x}$ and $J_{p,y}$ | $1.064 \times 10^{-9}$ and $8.718 \times 10^{-6}$ | $kg \cdot m^2$ |
| Coefficient of friction | $\mu$ | 0.3604 Identified (see Section 2.4.1) | |

### 2.4. System Architecture for Sim-to-Real Transfer to Solve the Geduldspiele Cubes

### 2.4.1. Identifying the Friction Coefficient $\mu$

A friction coefficient $\mu$ was marked as "identified" in Table 1. The idea to estimate it by system identification process is that (i) we apply the sequence of control to the real system and measure the trajectory of an iron ball, and (ii) we find $\mu$ which best reconstructs the measured trajectory of the iron ball under the simulation using (18). The system identification process to estimate $\mu$ is as follows:

1. Apply the sequence of motor command to a GS cube for 30 s, detect the position of the iron ball, and record those positions $(x, y)$ at every 200 ms.
2. Initialize $\mu$ by arbitrary value. Let $\Delta t = 20$ ms and run simulations using (21), and generate the trajectory of the ball. Extract every 10th $x$- and $y$-position of the ball and store them as $(x_s, y_s)$ (here, a subscript "s" is employed to distinguish the real measurement value from the simulation outputs).
3. Define error to be $e := \sum_{k=1}^{K} \|(x, y) - (x_s, y_s)\|_2$ and find $\mu^*$ that minimizes $e$.

By following the abovementioned process, we found $\mu^* = 0.3604$, which corresponded approximately to the friction coefficient of dry plastic–metal contact. When the sign(x) was substituted by tanh(10x) to guarantee continuity, $\mu^* = 0.0207$ and $\mu^* = 0.0213$ under different initial value setting, which was close to the friction coefficient of the lubricated plastic–metal contact [45].

### 2.4.2. Defining State, Action, and Algorithm to Virtually Train the Robotic Agent

Let $x_{rel}, y_{rel}$ be the relative position of an iron ball with respect to a hole position (also the goal of the geduldspiele cube problem) $(x_g, y_g)$, thus

$$x_{rel} := x - x_g \quad \text{and} \quad y_{rel} := y - y_g. \tag{22}$$

From (7), (10) and (11), we can obtain an insight to define a state vector: the state should involve the generalized coordinate elements as well as the linear and angular velocities along with them. For time step $t$, we define the state vector $s_t \in \mathbb{R}^{8 \times 1}$ to be

$$s_t = [x_{rel,t}, \dot{x}_t, \theta_{x,t}, \dot{\theta}_{x,t}, y_{rel,t}, \dot{y}_t, \theta_{y,t}, \dot{\theta}_{y,t}]^T, \tag{23}$$

Accordingly, the action $a_t \in \mathbb{R}^{2 \times 1}$ can be defined as

$$a_t = [\ddot{\theta}_x, \ddot{\theta}_y]^T \quad \text{where} \quad \ddot{\theta}_x, \ddot{\theta}_y \in [-0.5°, +0.5°], \tag{24}$$

which are related to two torques, $\tau_x$ and $\tau_y$, proportionally.

A reward $R_t$ at time step $t$ is defined by

$$R_t = e^{-c_1(x_{rel,t}^2 + y_{rel,t}^2)} - c_2(\theta_{x,t}^2 + \theta_{y,t}^2) - c_3(\ddot{\theta}_{x,t}^2 + \ddot{\theta}_{y,t}^2) \tag{25}$$

where $c_1, c_2, c_3$ are tuning parameters and were set to 0.001, 0.1, and 0.05, respectively.

To make the robotic agent learn an optimal policy $\pi^*(a_t|s_t)$, the proximal policy optimization (PPO) is adopted due to its well-known strength against the problems akin to control problems in continuous-time such as MuJoCo [46].

### 2.4.3. Our Sim-to-Real Transfer Architecture

Figure 4 shows the proposed sim-to-real architecture which consists of two main parts: a robotic manipulator and a geduldspiele cube in the real world, and their avatar system with the robotic agent in the simulation world. The key features of the proposed sim-to-real architecture are as follows:

- The robotic agent is pretrained with the geduldspiele cube of a flat plane under a reinforcement learning framework adopting PPO in the simulation world and knows $\pi^*(a_t|s_t)$.
- In the beginning, the state $s_t$ (measured by camera and mask-RCNN) and reward $R_t$ in/from the real world are sent to the simulator.
- Based on the received state and reward, the robotic agent in the simulation world generates the action $a_t$ to control the robotic manipulator.
- The robotic manipulator takes the action to the geduldspiele cube; as a result, $s_{t+1}$ and $R_{t+1}$ are sent to the simulator.
- Repeat the above process.

The details of the proposed sim-to-real architecture are discussed in Section 3 as well.

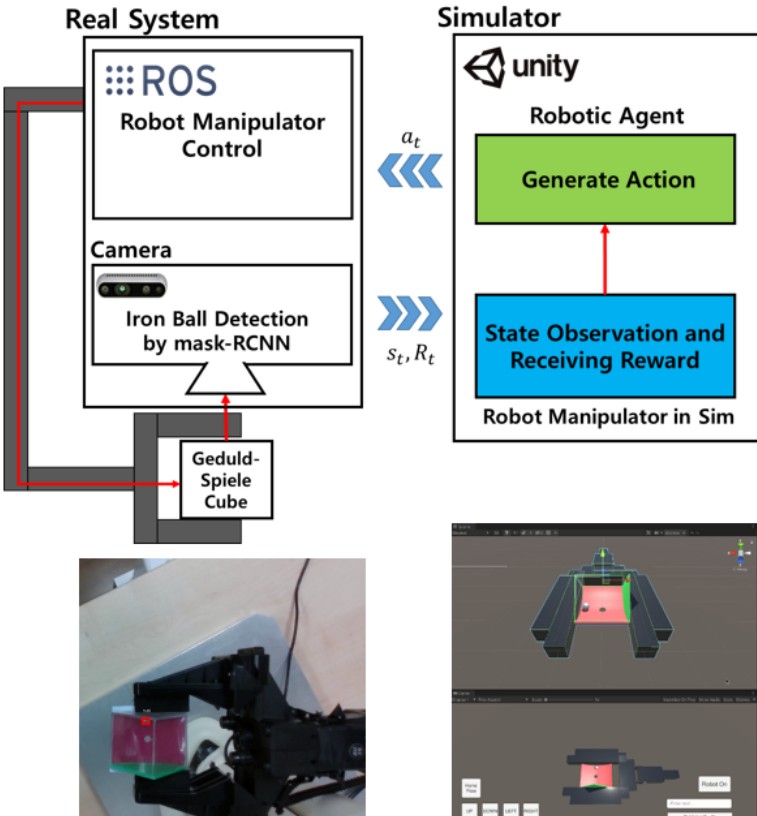

**Figure 4.** The proposed sim-to-real architecture: a real system diagram and camera view (**left**) and a simulator system diagram and virtual reality view (**right**).

## 3. Experiments

### 3.1. Experimental Setup

3.1.1. A Dual-Arm Robot, a Geduldspiele Cube, and a Camera in the Real World

Figure 5 shows our dual-arm robot, which will be referred to as geduldspiele (GS)-Bot throughout this paper. From Figure 5a, the GS-Bot has seven dynamixel actuators (XM430-W350-T, Robotis, Seoul, Korea): six for joints and one for a gripper per arm. It also has a pinch-type gripper in one hand, and an Intel RealSense camera (d435i, Intel, Hillsboro, OR, United States) is mounted on the other hand. All 14 dynamixel actuators are controlled via a controller + motor driver unit (U2D2 PHB, Robotis, Seoul, Korea) under ROS. The DH parameter for one arm of the GS-Bot is summarized in Table 2 (we note that the lateral direction from the shoulder is assumed to be $z_0$ of the base frame).

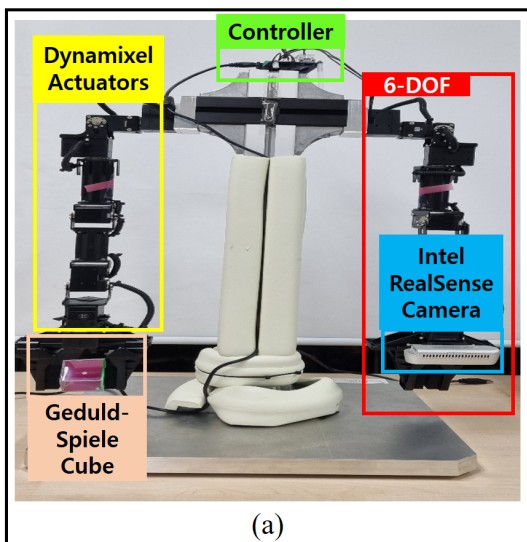
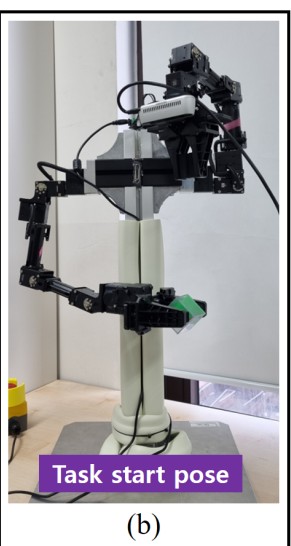

**Figure 5.** Our dual-arm robot manipulator. (**a**) A controller, dynamixel actuators, and Intel RealSense camera. (**b**) The dual-arm robot is ready to start solving the given geduldspiele cube.

**Table 2.** The DH parameter for one arm of our dual-arm robot.

| $i$ | $a_i$ | $\alpha_i$ | $d_i$ | $\theta_i$ | Remark |
|---|---|---|---|---|---|
| 1 | 0 | $-\pi/2$ | 77 | 0 | spherical shoulder #1 |
| 2 | 0 | $\pi/2$ | 0 | $\pi/2$ | spherical shoulder #2 |
| 3 | 0 | $-\pi/2$ | 164 | $\pi$ | upper arm |
| 4 | 24 | 0 | 0 | $-\pi/2$ | elbow |
| 5 | 124 | $\pi/2$ | 0 | $-\pi/2$ | spherical wrist #1 |
| 6 | 0 | 0 | $-120$ | 0 | spherical wrist #2 |

In Figure 5b, the GS-Bot is at a task start position from where the scene presented in Figure 4 (left bottom) can be taken by the camera from the top of the cube. At the this position, the $xy$-plane of the cube coordinate frame is aligned to be parallel to the ground. YOLO v4 was adopted to detect an iron ball inside a geduldspiele cube and identify the ball position to calculate the relative position with respect to the center hole. Then, the calculated relative position was Kalman-filtered. We note that only two actuators at the spherical wrist joint are used to manipulate the cube during the solving process.

### 3.1.2. A Virtual Wrist–Hand System and a Robotic Agent in the Simulation World

Figure 6 shows a virtual wrist–hand system in the simulation world according to three different cube types: flat, convex, and concave. All characteristics of this virtual wrist–hand system, e.g., scale, coordinate setting, etc., were matched to the real world system so that it worked as an avatar system in the simulation world for a robotic agent training and sim-to-real transfer.

For the training of the robotic agent, again, PPO was adopted under a reinforcement learning framework, which is supported by the Unity. The hyperparameters were set as follows: batch size = 10, buffer size = 100, learning rate = 0.0003, beta = 0.001, epsilon = 0.2, lambda = 0.99, num_epoch = 3, and learning_rate_schedule = linear. The definition of each hyperparameter can be found at [47].

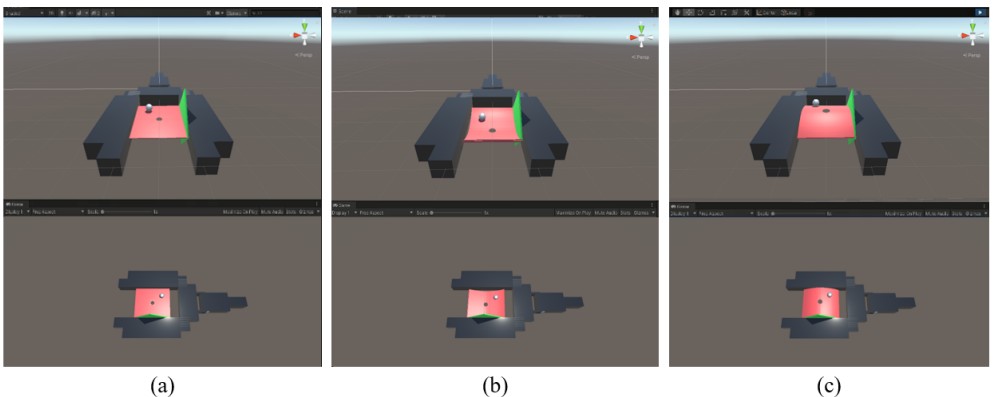

|          |          |          |
|:--------:|:--------:|:--------:|
|   (a)    |   (b)    |   (c)    |

**Figure 6.** Virtual wrist–hand system and a robotic agent: (**a**) a flat plane, (**b**) a convex plane, and (**c**) a concave plane. Two-DoF spherical wrist joint action is only used to solve the given geduldspiele cube.

### 3.1.3. Miscellaneous

We here summarize the miscellaneous remarks for our environmental setup as follows:

- To train YOLO v4, 13,000 iron ball images were used, including light reflections.
- The OS for the main desktop computer was Ubuntu 20.04.
- ROS release was ROS-noetic.
- Unity version was 2020.3.
- ROS and Unity send/receive data via TCP/IP.

### 3.2. Experiment 1: Training and Evaluating the Task Performance of the Robotic Agent for Solving the Virtual Geduldspiele Cubes in the Simulation Environment

In this experiment, we want to solve the problem of:

**Given:** The three "virtual" geduldspiele cubes.
**Train:** The robotic agent with the cube of a flat plane in the simulation world.
**Solve:** Each virtual geduldspiele cube in the simulation world.

The results will be analyzed in terms of success/fail and task completion time according to the three virtual geduldspiele cubes.

### 3.3. Experiment 2: Evaluating the Task Performance of the Trained Robotic Agent for Solving the Geduldspiele Cubes by Real Robotic Systems via Sim-to-Real Transfer

By performing this experiment, we want to scrutinize the feasibility of the proposed sim-to-real architecture for:

**Given:** The three "real" geduldspiele cubes.
**Employ:** The "virtually trained" robotic agent with the cube of a flat plane in the simulation world.
**Solve:** Each real geduldspiele cube in the real world.

Again, the results will be presented and compared in terms of success/fail and task completion time according to the three real geduldspiele cubes.

## 4. Results and Discussion

### 4.1. Result of Experiment 1

Figure 7 shows the snapshots while the robotic agent is solving the geduldspiele cubes in simulation world. The pictures in the left column depict the movement of the wrist–hand system along the time from top–bottom and left–right. The line graphs in the right column represent the relative distance from the center hole with time stamps corresponding to the snapshots on the left.

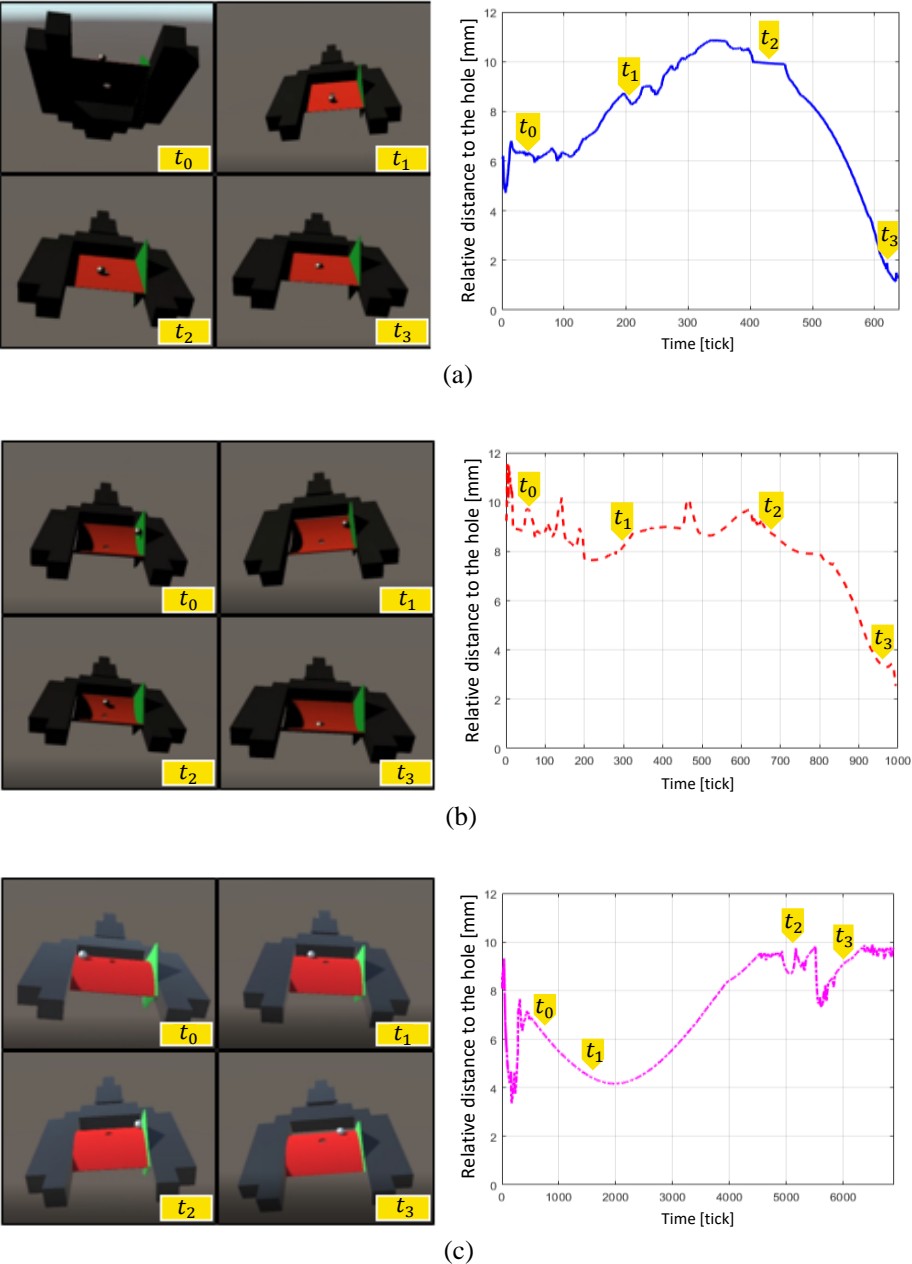

**Figure 7.** Snapshots (**left**) and corresponding relative distance to the hole (**right**) while the robotic agent is solving the geduldspiele cubes in the simulation world: (**a**) flat plane, (**b**) convex plane, and (**c**) concave plane. Time stamps are marked by $t_0$ through $t_3$. The flat cube and the convex cube are solved, but the concave is not.

From Figure 7, we can see that the robotic agent could succeed in solving the flat plane cube and the convex plane cube, but could not for the concave one. Figure 8 shows that the convex one showed the best performance in terms of task completion time in the simulation world.

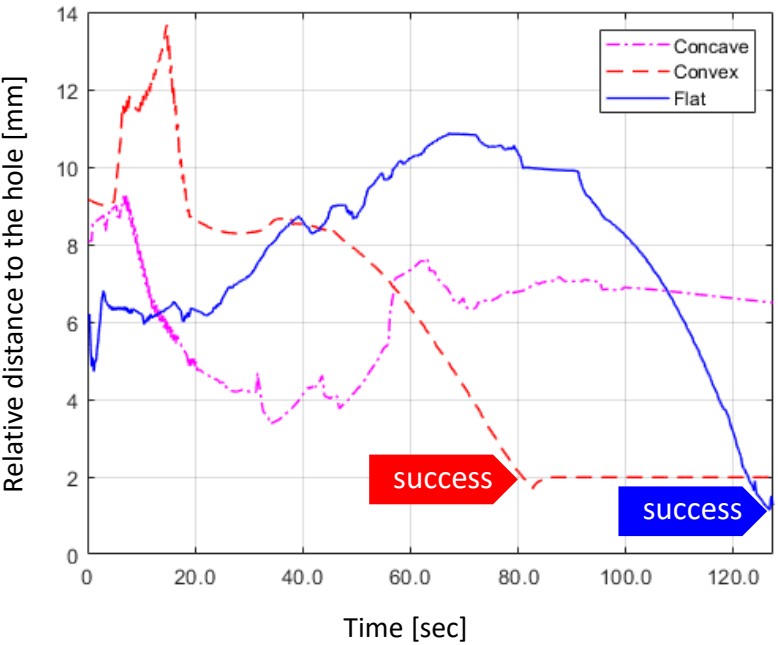

**Figure 8.** The comparison result of the trained robotic agent performance given the virtual GS cubes of the flat plane, convex plane, and the concave plane in terms of success/failure and task completion time in the simulation world.

## 4.2. Result of Experiment 2

Figure 9 shows the sim- and the real-world scenes as well as the relative distance while the trained robotic agent is solving the real geduldspiele cubes. For a flat plane and a convex plane, as shown in Figure 9a,b, the real geduldspiele cube problems were solved via our sim-to-real transfer architecture. We note that since the unit of the relative distance is mm, there exists a small error boundary, which is caused by state observation (thus, camera measurement error), even after the iron ball reached the center hole.

Again, in the case of the real geduldspiele cube problem, the trained robotic agent failed to solve the concave one. The failure cases of the concave cube in both the virtual and the real world will be discussed in detail in the following section. Figure 10 presents the comparison results of the trained robotic agent performance given the real cubes of the flat plane, convex plane, and the concave plane in terms of success/failure and task completion time. For more information, please refer to Video S1.

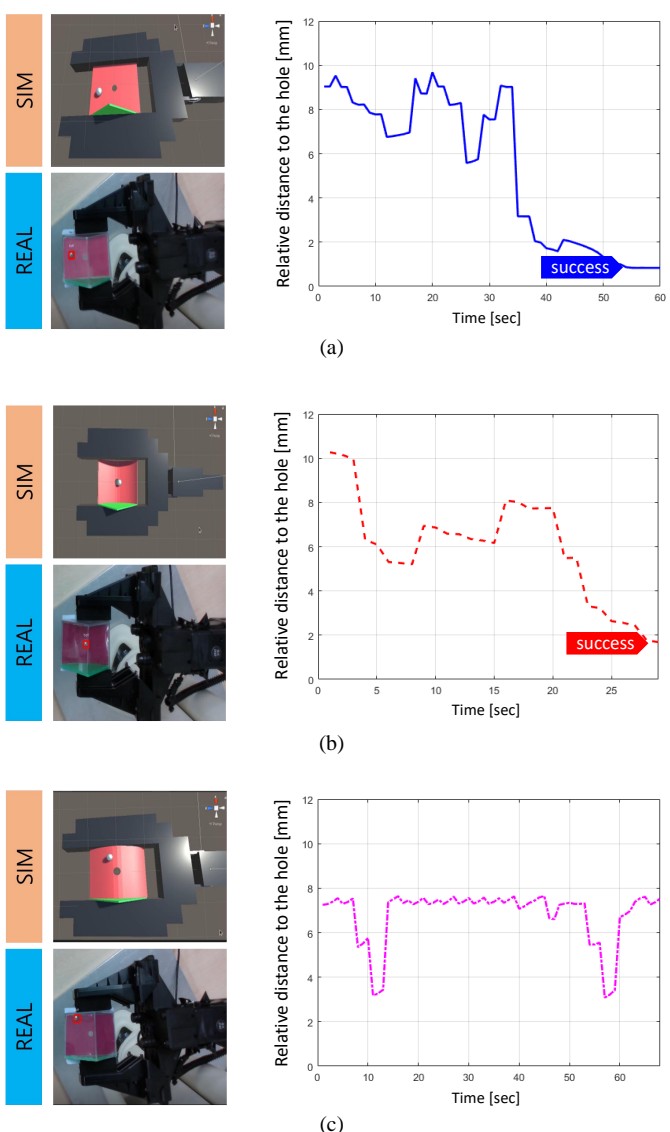

**Figure 9.** The sim- and the real-world scenes as well as the relative distance while the trained robotic agent is solving the real geduldspiele cubes: (**a**) flat plane, (**b**) convex plane, and (**c**) concave plane.

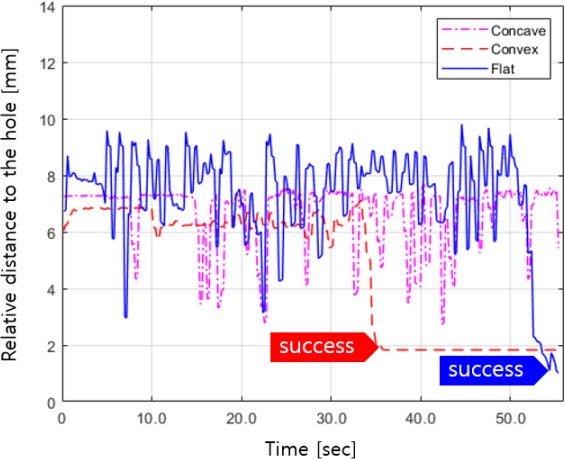

**Figure 10.** The comparison results of the trained robotic agent performance given the real GS cubes of the flat plane, convex plane, and the concave plane in terms of success/failure and task completion time.

*4.3. Discussion*

When the proposed approach was initiated, we expected that the robotic agent trained with a flat curve would be able to solve the cubes of different curvatures. This expectation had a basis on the existing findings in [39] and the strength of PPO for continuous-like control problems, e.g., mountain car, MuJoCo, etc. However, in our sim-to-real framework, the trained robotic agent failed to solve the concave cubes in both the virtual and the real world.

One possible reason is the discrepancy of the problem itself between the mountain car problem and the concave geduldspiele cube, especially for the termination condition. In the mountain car problem, merely passing the goal is accounted for as a success, although the car does not stay at the goal. In contrast, for the concave cube problem, the reinforcement learner should learn the optimal policy to reach the goal as well as to maintain the ball around the goal.

Another possible reason is that the concave cube problem seems similar to flat and convex cube problems geometrically; however, the required level of the dexterity for force control might be truly difficult. In this case, a meta-learning-based approach would be a promising solution.

Lastly, the limitations of this study can be summarized as follows:

- The geduldspiele cubes considered in this study were simpler ones. Generally, the other cubes are much more difficult to solve.
- We set the sampling time for both the sending/receiving $(s_t, R_t)$ and $a_t$ via TCP/IP small enough; however, there might exist a delay which can affect the proposed system performance since it is not indeed real time (or even close to real time).

**5. Conclusions and Future Work**

This paper proposed an approach to solve the three GS cubes via sim-to-real transfer. We started by deriving the dynamic models of ball–plane and ball–hole. We then presented the continuous and discrete state-space model. The optimization-based approach to identify a friction coefficient was introduced. We also presented the definition of a state as well as reward function to formulate the RL problem, followed by showing our sim-to-real transfer architecture. From the experimental result, we could find the answer for our research question; that is, the virtually trained robotic agent was able to solve the geduldspiele cubes of a flat plane as well as a convex plane. However, the concave was solved neither in the real world nor in the virtual world.

The contributions of this paper can be summarized as follows. First, we derived the dynamic model for a simple GS cube and presented the LTI system model. Next, the optimization-based approach to estimate a friction coefficient between ball and plate was introduced. The sim-to-real transfer architecture to solve the GS cube was proposed; finally, the results showed that the flat and convex planes were solved under the proposed approach. Therefore, this study substantiates that the optimal policy to imitate human-like behavior could be obtained by applying the proposed approach.

Future work should be followed in the direction of elaborating the proposed approach and challenging more complex GS cube problems.

**Supplementary Materials:** The following supporting information can be downloaded at: https://www.mdpi.com/article/10.3390/app121910124/s1, Video S1: Can a Robotic Agent Solve Simple Geduldspiele Cubes via Sim-to-Real Transfer?

**Author Contributions:** Conceptualization: H.-U.Y.; methodology: J.-H.Y. and H.-U.Y.; software: J.-H.Y., H.-J.J., J.-H.K., D.-H.S. and H.-U.Y.; formal analysis: J.-H.Y., J.-H.K. and H.-U.Y.; investigation: J.-H.Y., H.-J.J. and H.-U.Y.; resources: D.-H.S. and H.-U.Y.; data curation: J.-H.Y. and H.-U.Y.; writing—original draft preparation: J.-H.Y., H.-J.J., J.-H.K., D.-H.S. and H.-U.Y.; writing—review and editing: J.-H.Y., H.-J.J., J.-H.K., D.-H.S. and H.-U.Y.; visualization: J.-H.Y., H.-J.J. and H.-U.Y.; supervision: H.-U.Y.; project administration: H.-U.Y.; funding acquisition: H.-U.Y. All authors have read and agreed to the published version of the manuscript.

**Funding:** This work was supported by the National Research Foundation of Korea (NRF) grant funded by the Korea government (MSIT) (Grant No. 2021R1F1A1063339) and the MSIT National Program for "Excellence in SW (Grant No. 2019-0-01219)" supervised by the Institute of Information and Communications Technology Planning and Evaluation (IITP) in 2022.

**Institutional Review Board Statement:** Not applicable.

**Informed Consent Statement:** Not applicable.

**Data Availability Statement:** Not applicable.

**Conflicts of Interest:** The authors declare no conflict of interest.

## Abbreviations

The following abbreviations are used in this manuscript:

| | |
|---|---|
| RL | Reinforcement learning |
| ML | Machine learning |
| GS | Geduldspiele |
| LTI | Linear time invariant |
| PPO | Proximal policy optimization |
| ROS | Robotic Operating System |

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
