# Peer review of "Solving a Simple Geduldspiele Cube with a Robotic Gripper via Sim-to-Real Transfer"

_applsci, doi:10.3390/app121910124_

Round 1

Reviewer 1 Report

There is two point of view in this manuscript:

1-      In scientific view your method is the same as inverted pendulum. If there is any significant scientific difference between inverted pendulum control methods and your idea in your proposed method, let us know it.

2-      In practical view this is an excellent piece of work in which the theoretical things bring to practice.

Author Response

Please refer to the attached author's response. Thanks for your great regards.

Reviewer 2 Report

1) The authors need to refine the title of this paper

2) The introduction section needs to be enhanced through the incorporation of more and relevant information in the problem domain.

3) How does this paper extend the state of the art approaches in this domain?

4) The related work section should be included

5) There is need for comparative analysis of the proposed approach.

6) The conclusion section needs to be improved

7) More recent and relevant references need to be incorporated in this paper.

Author Response

(The authors gave the same response as above.)

Reviewer 3 Report

usually for an journal paper at least 50% of references are usually recommended as for the rest the paper is safe and sound

Author Response

(The authors gave the same response as above.)

Reviewer 4 Report

The authors introduced a manuscript titled “Can a Robotic Agent Solve Simple Geduld-Spiele Cubes via Sim-to-Real Transfer?”. The paper is interesting and well-explained. This is an excellent manuscript from the authors. I would recommend the editors accept the manuscript with some of minor corrections as in the following:

1-      The title is unwanted, I suggest the title to be modified without question form.

2-      What are the new features and advantages of the proposed approach when compared with other similar approaches? That must be highlighted in the conclusions.

3-      The proposed approach should be presented by summary flowcharts only.

4-      The accuracy of some figures is low (for example Fig.6,7, and 8).

5-      What numerical techniques were used in the whole article investigation (especially for optimization technique); this can be inserted in the article?

6-      The conclusions are very abbreviated, the achieved advantages of the approach can be inserted.

7-      Can the disturbances can be considered in the torque evaluations? Please show that.

8- The stability of the controllers is not guaranteed.

9-      The authors should be cited more recent references.

Author Response

(The authors gave the same response as above.)

Round 2

Reviewer 2 Report

The paper has been greatly improved. However, the authors need to consider the format of the conclusion section.

Author Response

Thanks for your valuable suggestion. The format of conclusion part has been revised as follows:

"The contributions of this paper can be summarized as follows. First, we derived the dynamic model for simple GS-cube and presented the LTI system model. Nest, the optimization-based approach to estimate a friction coefficient between ball and plate was introduced. The sim-to-real transfer architecture to solve GS-cube was proposed; finally, the results showed that flat and the convex planes were solved under the proposed approach. Therefore, this study substantiated that the optimal policy to imitate human-like behavior could be obtained by applying the proposed approach. Future work should be followed in the direction of elaborating the proposed approach and challenging more complex GS-cube problems."

Please refer the line 289 through 297 in the revised manuscript.